# Hell’s Gate Globin-I from *Methylacidiphilum infernorum* Displays a Unique Temperature-Independent pH Sensing Mechanism Utililized a Lipid-Induced Conformational Change

**DOI:** 10.3390/ijms25126794

**Published:** 2024-06-20

**Authors:** Brandon J. Reeder, Dimitri A. Svistunenko, Michael T. Wilson

**Affiliations:** School of Biological Sciences, University of Essex, Wivenhoe Park Colchester, Essex CO4 3SQ, UK; svist@essex.ac.uk (D.A.S.); wilsmt@essex.ac.uk (M.T.W.)

**Keywords:** hemoglobin, lipid binding, pH dependence, heme coordination, pH sensing, extremophile

## Abstract

Hell’s Gate globin-I (HGb-I) is a thermally stable globin from the aerobic methanotroph *Methylacidiphilium infernorum*. Here we report that HGb-I interacts with lipids stoichiometrically to induce structural changes in the heme pocket, changing the heme iron distal ligation coordination from hexacoordinate to pentacoordinate. Such changes in heme geometry have only been previously reported for cytochrome c and cytoglobin, linked to apoptosis regulation and enhanced lipid peroxidation activity, respectively. However, unlike cytoglobin and cytochrome c, the heme iron of HGb-I is altered by lipids in ferrous as well as ferric oxidation states. The apparent affinity for lipids in this thermally stable globin is highly pH-dependent but essentially temperature-independent within the range of 20–60 °C. We propose a mechanism to explain these observations, in which lipid binding and stability of the distal endogenous ligand are juxtaposed as a function of temperature. Additionally, we propose that these coupled equilibria may constitute a mechanism through which this acidophilic thermophile senses the pH of its environment.

## 1. Introduction

Hemoglobins (Hbs) are a family of proteins with a heme moiety that serve different functions depending on their heme pocket geometry, having different affinities for gaseous ligands [1]. Hbs can function as oxygen carriers or perform redox reactions, including acting as nitric oxide (NO) scavengers or NO generators, and be involved in hypoxic stress response mechanisms [2]. Globins of prokaryotic organisms are typically truncated compared to their vertebrate relatives, serving a number of functions including oxygen and NO chemistry [3]. Prokaryotes also have Hbs that have a high sequence homology and structurally very similar to vertebrate globins [4]. Hell’s Gate globin-I (HGb-I) is one of five globins from the acidophilic thermophilic aerobic methanotroph *Methylacidiphilium infernorum* [5] (initially named ‘*Methylokorus infernorum*’). Revealed from the genome of *M. infernorum* [6], HGb-I is named after the region of geothermal vents where the extremophile is found (Hell’s Gate, Tikitere, New Zealand) [7]. Of the encoded globin proteins, only globins I and IV have been examined by recombinant protein synthesis to date [5,8]. The structural differences between HGb-I and HGb-IV are considerable: HGb-IV is a truncated two-on-two α helical globin [5] typical of most prokaryotic globins, whereas HGb-I is more similar to that of many three-on-three α helical vertebrate globins [8]. This suggests that the globins of *M. infernorum* serve distinct functional roles [9].

Homology modelling of HGb-I suggests that the protein closely resembles human neuroglobin (Ngb) with a sequence homology of >33% [5]. Other vertebrate globins also have some sequence homology varying between human hemoglobin β chain (29%) and human cytoglobin (Cygb) at 21% sequence homology. HGb-I exhibits high O_2_ affinity and low rates of autoxidation in the temperature range of 20–50 °C [5]. The optical spectrum of the deoxygenated ferrous protein shows typical properties of a hexacoordinate protein at neutral and alkaline conditions, similar to vertebrate globins Ngb and Cygb. However, under acidic conditions HGb-I is primarily pentacoordinate [5]. Many invertebrate and bacterial globins, including HGb-I, have a distal E7 position occupied by a glutamine residue [10]. With vertebrate globins, this distal position is normally occupied by a histidine residue, with notable exceptions including myoglobin (Mb) from African and Asian elephants that also have a distal glutamine [11] as well as human androglobin [12].

A tyrosine is in the B10 position in HGb-I, common in many bacterial truncated globins and some nematodes (e.g., *Ascaris suum*) [13]. This tyrosine functions to stabilize ligand binding through hydrogen bonding additional to that provided by the distal ligand [14,15]. The current crystal structures available for HGb-I (O_2_ and acetate bound) supports the hydrogen bonding to the distal ligand from both E7 glutamine (Gln50) and B10 tyrosine (Tyr29) [8]. However, observations from the current crystal structures show that neither the Gln50 nor Tyr29 can coordinate to the heme iron, even allowing for some sliding of the heme [5]. Molecular dynamic simulations have suggested that the conformational movements of the B10 and E7 residues, as well as structural flexibility of the GH loop and H-helix, are involved in modulating the ligand binding properties of the protein [5]. Additionally, simulations also suggest a ligand migration pathway resulting in net slower de-oxygenation compared to other globins [16]. HGb-I exhibits low electrochemical potential with an unusual binding of ligands to the ferrous protein normally only observed with ferric globins [17].

Previous studies on Cygb have shown that the protein distal histidine–iron ligation in the ferric form of the protein is labile, changing from hexa- to pentacoordination upon binding of molecules such as lipids and free fatty acids [18]. This is dependent on the presence of an intramolecular disulfide bond between the B and E helices, enhancing the distal histidine off rate by six hundred-fold compared to protein without the intramolecular disulfide bond [19]. There is a similar effect with the interaction of cytochrome c and cardiolipin, although the lipid to protein ratio required to induce this change is much greater compared to Cygb [20,21,22]. Here we report that HGb-I exhibits a clear interaction with lipids, which induce structural changes from hexacoordinate to pentacoordinate under specific pH ranges, similar to that observed for human Cygb. The stoichiometry and dissociation constant of lipid binding to HGb-I are essentially identical to those of Cygb, suggesting that HGb-I, although more structurally homologous to Ngb than Cygb, has potential functional aspects more related to Cygb. HGb-I shows lipid-induced effects on heme pocket structure in both ferric and ferrous oxidation states, unlike Cygb and cytochrome c, which only show effects in the ferric oxidation state [18,23]. The changes in heme iron coordination are pH-dependent, but temperature-independent. We propose a mechanism to explain these observations, in which the initial rapid, high-affinity lipid binding does not directly result in a change in heme iron coordination, but shifts the iron coordination equilibrium, resulting in a slow lipid concentration-independent change in coordination state. Furthermore, an increase in temperature decreases the stability of the hexacoordinate species, but is offset by a lower affinity of lipid binding.

HGb-I exhibits no lipid peroxidase activity, ruling out lipid-based signaling as a response to oxidative stress proposed for Cygb [18]. Nonetheless, the ability of HGb-I to respond to both lipid and pH changes, independent of temperature, by alteration of the heme iron coordination remains an intriguing characteristic of this extremophile’s globin.

## 2. Results

### 2.1. Optical Properties and Extinction Coefficients of HGb-I

Figure 1 shows the optical spectra of recombinant HGb-I. Extinction coefficients at pH 7.4 were measured by comparison of optical spectra with heme concentration measured by HPLC described in the Section 4. This technique follows the same principle as the historically more commonly used hemochromagen assay [24,25]. The HPLC and hemochromagen methods both dissociate the heme from the protein through acid-induced breakage of the acid-labile link to the proximal histidine. The heme is then isolated in a volatile organic solvent (here acidified acetonitrile instead of acidified pyridine) for comparison with samples of known concentration. The Soret peak and visible peak wavelengths and coefficients for the ferric, deoxyferrous and ferrous-CO forms of the protein can then be calculated and are given in Table 1.

### 2.2. The pH Dependence of the Heme Coordination of HGb-I

The optical spectrum of ferric HGb-I at pH 7.4 is typical of hexacoordinate heme ligation (see Figure 1) and is similar to other hexacoordinate ferric globins such as Cygb [26] and Ngb [27]. On lowering the pH, this spectrum changes; the Soret undergoes a hypsochromic shift from 412 to 407 nm, the α and β bands of the visible region decrease in amplitude and a peak appears at ~620 nm (this is identical to the optical changes seen upon lipid addition, *vide infra*). These changes are typical of a switch in heme coordination from primarily low-spin (LS) hexacoordinate to high-spin (HS) pentacoordinate on lowering pH, as previously observed with ferric Cygb [18,28]. Both ferric and ferrous HGb-I exhibits an acid–alkaline transition between hexacoordinate heme iron ligation state at alkaline pH and pentacoordinate state at more acidic pH values (Figure 2A,B, ●). Changes in optical spectra for the deoxyferrous and ferric protein have been reported previously, showing a general change from LS to HS reported for ferric and deoxyferrous HGb I as pH decreases [5]. Here we have determined the pK of these transitions, with the ferric protein showing an acid–alkaline transition with pK_a_ of 5.50 ± 0.05 and the ferrous protein a pK_a_ of 8.43 ± 0.06. There is an additional pK_a_ with the ferrous protein exhibiting minor changes in optical intensity at pH 5.79 ± 0.42.

The kinetics of the pH transition in the ferric and ferrous protein were investigated by pH jump experiments. In these experiments, the ferric protein at pH 10 was mixed with lower pH buffer and the kinetics of the formation of a predominantly HS pentacoordinate species monitored by stopped flow spectroscopy. The time courses can be described by single exponentials, and the dependence of the observed rate constant (k_obs_) on pH (final) is shown in Figure 3. From this figure it can be seen that k increases as the pH decreases, and the data fit to a single proton transition, with pKa of 3.00 ± 0.07 for the ferric protein and 2.10 ± 0.12 for the ferrous protein. This dependence, together with the titration data, allow us to construct a model for the pH-dependent coordination change in the ferric protein (see Discussion). At higher pH values, there is a pK_a_ for both ferric and ferrous ~pH 7–7.6 (Figure 3B) with an additional pK_a_ ~4.5–5 for the ferrous protein. The effects of these on the kinetics are very small in comparison to the kinetic changes observed at below pH 5. Furthermore, the amplitude changes are also very small (Appendix A). It is probable that these represent the effects of a protonation of one of the amino acid residues close to the heme iron.

The optical changes observed for ferric and ferrous HGb-I during the pH jump at a low pH are compared in Appendix A. For both ferric and ferrous protein, the initial spectrum following the pH jump (~1.2 ms) showed the hexacoordinate species. At pH 3.5 and above (Appendix A), the transition is to the pentacoordinate form. At pH values below 3.5, it is somewhat obscured by spectral changes associated with denaturation of the protein (Appendix A), probably triggered by protonation of the proximal histidine 26. However, the transition to the penta form is clearly discerned at pH 2.5 but incomplete before significant denaturation (Appendix A). The kinetics that are reported in Figure 3A are those of transition from the hexacoordinate to pentacoordinate form.

### 2.3. Spectral Changes Consequent on Lipid Binding

The addition of lipids, such as the fatty acid oleate, to the ferric protein-induced changes in the optical spectrum of HGb-I, essentially identical to those seen on lowering the pH, indicating a change in the coordination state of the iron from hexa- to pentacoordination (Figure 4A,B).

The ferrous protein also shows optical changes when oleate is added (Figure 4C,D). Again, this is typical of a change in heme iron coordination from hexacoordinate to pentacoordinate, with essentially identical optical changes as shown upon acidification of the protein [5]. However, the changes in the ferrous protein do not appear to be complete, and even high oleate concentrations do not fully convert the protein to the HS form. Nonetheless, this lipid-induced effect on the heme iron coordination in deoxy-ferrous HGb-1 contrasts with deoxy-ferrous Cygb, where no change in the ferrous heme iron coordination was observed [18].

### 2.4. EPR Confirmation of Spin State Change

To confirm the optical assignment, the EPR spectra of the ferric protein in the absence and presence of oleate were taken (Figure 5). In both samples, a mixture of two distinct species representing two spin states of the ferric iron were observed. One species is characterized by an axial symmetry EPR signal with g_⊥_ = 5.91 and g‖ = 2.00. This signal is indicative of a HS ferric heme state, showing the protein iron in a pentacoordinate ligation geometry, typically HisF8-Fe(III)—H_2_O [29,30]. The second species is a LS ferric heme state with an EPR signal usually comprising three components [29] but when it is a highly anisotropic LS form, with a g_z_ > 3, often the outmost high field g_x_ component has a comparably low value [31] and, according to the Aasa & Vanngard 1975 ‘re-examination’, should have a comparably lower intensity [32]. Indeed, for the LS heme signal with g_z_ > 3, the third component is often undetectable, e.g., [33]. The second component of the EPR spectrum of the LS ferric heme form is likely overlaid with the g = 2.05 signal (mainly originating from adventitious Cu^2+^ ions), having slightly greater than 2.05 g-value (Figure 5B). The LS ferric heme signal is indicative of the heme iron in a hexacoordinate geometry, consistent with a HisF8-Fe(III)-GlnE7 or LysE10 ligation. Ferric Cygb exhibits similar signals at g = 5.87 and g = 2.00 for HS (HisF8-Fe(III)—H_2_O) and g = 3.18 and g = 2.05 for LS (HisF8-Fe(III)-HisE7) [18].

At pH 7.4 and in the absence of sodium oleate, the LS signal is dominant and responsible for 91% of the protein concentration with 9% of the protein in the HS state. In the presence of 3-fold excess oleate, the proportion is reversed—the HS signal becomes dominant, accounting for ~90% of overall EPR detectable ferric heme. This compares to a shift of 70% LS Cygb in the absence of oleate to 25% in the presence under identical conditions [18]. Thus, like human Cygb, EPR spectroscopy confirms that HGb-I heme iron coordination shifts from a predominantly hexacoordinate to primarily pentacoordinate form in the presence of oleate. It should be noted that the process of freezing can create artefacts due to changes in pH and alteration of the dynamics of the spin states as previously observed with Cygb [18]. However, the spin states observed by EPR match reasonably well the optical data in Figure 2 and Figure 4.

### 2.5. Lipid Binding Titrations

The normalized amplitude of the absorbance changes seen in Figure 4, which accompany the ferric heme coordination change in HGb-I (400–416 nm, pH 7.0), were plotted as a function of lipid concentration (Figure 6) and were fitted using Equation (7). From the fit, the apparent dissociation constant (K_L(obs)_) of the oleate at 20 °C (Figure 6A) was determined as 0.73 µM ± 0.03 µM, essentially identical to that of human Cygb under matching conditions [18]. Surprisingly, the effect of higher temperatures on the lipid binding was minimal, with a dissociation constant of 1.03 µM ± 0.10 µM at 50 °C. This behavior will be rationalized in the discussion below. The stoichiometry of protein:lipid binding was close to 1:1.3 at 20 °C and 1:1.7 at 50 °C (6.98 µM ± 0.25 µM and 8.82 µM ± 0.10 µM lipid for the 5 µM protein at 20 and 50 °C, respectively). HGb-I was also titrated with sodium palmitate (Figure 4A). Optical changes for palmitate binding to HGb-I were identical to those observed for oleate. Palmitate, being a saturated fatty acid, is ~50 times less soluble in water compared to the monounsaturated sodium oleate. However, at 50 °C, the solubility was sufficient for titration. The stoichiometry of protein:lipid binding was approximately equal to that of the oleate binding, but with a larger K_L(obs)_ value of 9.88 µM ± 0.39 µM.

The deoxyferrous protein exhibited a titration curve that was also fitted to Equation (7) (440–427 nm, Figure 6B). Here the apparent stoichiometry of the protein:lipid was much less at, 1:0.25 (1.27 µM ± 0.064 µM and 1.14 µM ± 0.084 µM lipid for the 5 µM protein at 20 and 50 °C, respectively). However, the inspection of Figure 2B shows that at pH 8.5, the pH where the maximum change in coordination is observed, only a fraction of the protein participates in the LS to HS transition. This fraction is estimated to be ~1.6 µM protein of the total 5 µM protein concentration. This indicates that the true stoichiometry of binding is close to 1:1. There is an additional optical change at higher oleate concentrations observed at 50 °C (fitted to straight line, Figure 6B). The optical changes are distinct from the change in heme iron coordination and may represent partial unfolding of the protein as has been observed with Cygb and Ngb at high oleate concentrations [18]. The K_L(obs)_ values for the oleate binding are similar at the two temperatures; 0.21 µM ± 0.018 µM (20 °C) and 0.31 µM ± 0.020 µM (50 °C). Palmitate also induces an optical change that, like the ferric protein, exhibits a similar stoichiometry to the oleate binding, but has a higher K_L(obs)_ (3.43 µM ± 0.13 µM). With both ferric and ferrous protein, the K_L(obs)_ for palmitate is ~10–15 times higher than observed with oleate.

The addition of sodium oleate (five-fold excess) shifted the pK_a(obs)_ of the LS to HS transition (described above) to higher values i.e., oleate stabilizes the pentacoordinate HS form (Figure 2A,B, ■). Oleate (25 uM) shifts the acid–alkaline transition pK_a(obs)_ to 8.40 ± 0.04 and 9.24 ± 0.09 for ferric and ferrous protein, respectively, an alkaline shift of 2.90 and 0.89 pH units, respectively. The ferric protein in the presence of oleate shows an additional minor pK_a(obs)_ at 5.92 ± 0.58, possibly the same protonation event as observed with ferrous protein in the absence of oleate. This minor transition is accompanied by minimal changes in optical spectra and likely arise from protonation of a group in the proximity of the iron, or changes in hydrogen bonding. The oleate has a maximum effect on heme iron ligation at pH values ~7.0 and ~8.5 for ferric and ferrous HGb-I, respectively (Figure 2C).

### 2.6. Kinetics of Optical Changes of Lipid Binding to HGb-I

The kinetics of the optical changes resulting from the change in heme iron coordination for HGb-I were measured as a function of oleate concentration for ferric and ferrous protein (Figure 7). The observed rate constants for the LS to HS transition were independent of lipid concentration over the concentration range observed. For ferric HGb-I (Figure 7A), the observed values were 2.32 × 10^−2^ ± 3.3 × 10^−3^ s^−1^ and 7.76 × 10^−4^ ± 2.1 × 10^−4^ s^−1^ for the fast and slow phases, respectively. The slow phase was dominant, comprising ~85% of the optical change at this wavelength; thus for the dominant slow fraction, optical changes were 10^3^–10^4^ fold slower than those observed with Cygb over the concentration range studied. For ferrous HGb-I (Figure 7B), the kinetics were also biphasic, and the fast and slow phases were both lipid concentration-independent. However, the optical changes in the ferrous protein were an order of magnitude faster (2.37 × 10^−1^ ± 1.7 × 10^−2^ s^−1^ and 3.67 × 10^−2^ ± 9.92 × 10^−3^ s^−1^) when compared to the ferric protein. Additionally, the fast phase for the ferrous protein is dominant, responsible for ~80–90% of the amplitude change.

### 2.7. Effects of pH and Oleate on Ligand Binding

The effect of oleate on ligand binding to HGb-I was examined. Appendix A shows the effect of oleate on the binding of azide to ferric HGb-I. Here, the presence of oleate enhanced azide binding at pH values of 6 and above. Below pH 6, the presence of oleate inhibited azide binding. Azide has a pK_a_ of 4.6; the charge of the azide molecule likely affects binding at a low pH, with the protonated form (HN_3_) significantly increasing the binding rates to HGb-I, but only in the absence of oleate. The amplitude change (Appendix A) shows pK_a_ of 5.8 and 6.3 in absence and presence of oleate, respectively, with a small secondary pK_a_ at 8.3 for the oleate bound protein.

Appendix A shows the effect of pH and oleate on the kinetics of CO binding to ferrous HGb-I. In the absence of oleate, the CO binding to the hexacoordinate form is slow and reaches a plateau, as would be expected from the dissociation from the intrinsic ligand. At pH 5, however, where the pentacoordinate form dominates, the CO binding kinetics could not be observed on the timescale of the stopped flow, even at concentrations of CO as low as 12.5 µM. This is to be expected, as the binding of CO to the pentacoordinate form, as seen by flash photolysis, is >3.5 × 10^7^ M^−1^s^−1^ [5]. In the presence of oleate, a significant increase was observed in the rate constant for CO binding (three-fold at pH 10 and over six-fold at pH 8.5), reflecting enhanced k_off_ values for the intrinsic ligand. The K_D_ for CO binding at pH 10 and 8.5 was not significantly changed by the presence of oleate in this pH range.

### 2.8. Lipid Peroxidase Activity of HGb-I

We have previously proposed that the lipid-induced changes in heme iron coordination of Cygb could be linked to the high peroxidase activity of the protein, resulting in generation of oxidized lipid-based molecules with potential cell signaling properties under conditions of oxidative stress [18,28]. Other heme proteins such as Mb have been shown to induce the formation of potent vasoactive compounds such as isoprostanes under oxidative conditions, resulting in pathological consequences such as acute kidney injury following rhabdomyolysis [34,35,36]. For Cygb, this mechanism of lipid oxidation would not be pathological due to the low concentrations of protein within the cell but rather would be a mechanism for cell signaling. Here we examined the lipid peroxidase activity of HGb I and compared this to Mb and Cygb. As shown in Figure 8A, both Cygb and Mb oxidize the liposomes within a few hours, resulting in the net formation of lipid-based conjugated dienes measured at 234 nm. The mechanism of lipid oxidation is complex, but small amounts of lipid peroxides within the lipid bilayer react with the protein to generate the ferryl oxidation state. This ferryl heme iron in turn reacts with lipids to generate more lipid peroxides. This ‘lag period’ proceeds until a cascade of lipid oxidation occurs, seen by the rapid increase on conjugated dienes [2,37]. The profile of the oxidation of liposomes by Mb and Cygb are essentially identical to those previously reported, with lag periods in the range of a few minutes and high maximal rates of lipid oxidation of 44.1 ± 6.25 nM s^−1^ and 8.36 ± 2.26 nM s^−1^ for Cygb and Mb, respectively [2,18,28,37]. The rates of HGb-I oxidation of liposomes are negligible compared to Mb and Cygb, with no cascade oxidation event observed within the timeframe of the experiment (Figure 8A). Thus HGb-I shows no significant ability to generate lipid oxidation products (1.07 ± 0.256 nM s^−1^) and is only marginally higher than the autoxidation of lipids observed in the controls (0.50 ± 0.215 nM s^−1^).

## 3. Discussion

The effect of oleate on the sixth coordination site of the heme iron is significant for both ferric and ferrous oxidation states. This contrasts with Cygb, where only the ferric form of the protein is affected by oleate binding [18], but may be related to the unusual ferrous ligand binding observed with HGb-I and ligands like cyanide and imidazole that are more readily observed binding to ferric heme proteins [17]. Additionally, the lipid binding to ferric Cygb does not cause a shift in the pK for the acid–alkaline transition (His-Fe(III)-His to His-Fe(III)—H_2_O) but eliminates the His-Fe(III)-His form at an acid pH. Nonetheless, the maximum effect of oleate on the coordination state of ferric Cygb, like HGb-I, is also ~pH 7 [18]. A hexacoordinate, LS conformation, has been previously noted for other hemoglobins with a distal E7-Gln and B10-Tyr such as of the hemoglobin of the cyanobacterium *Synechocystis PCC6803* [38] and *Chlamydomonas eugametos* [39]; see Appendix A for sequence alignment comparison. However, crystal structures of *Synechocystis* Hb show that the distal ligand is occupied by a histidine originating from the E10 helical position [40,41]. A lysine residue, also from the E10 position, is also noted on similar truncated Hbs such as *C. eugametos*, *P. caudatum* and *M. tuberculosis* [41]; see Appendix A. *M. infernorum* also has an HGb-I like domain attached to a Roadblock/LC7 domain, known as HGbRL 38. This protein, unique to the genus *Methylacidiphilum*, may have evolved from duplication of the HGb-I and Roadblock/LC7 proteins. The crystal structure shows a dimeric protein with a site-swapped open and closed form of the distal heme pocket. The closed form of the dimeric protein shows the distal site occupied by the E10 Lys [42], suggesting that this is a possible ligand in the monomeric HGb-I.

The *C. eugametos* Hb shows similar pH dependencies of ferrous heme iron coordination compared to HGb-I, although two major transitions are observed. The two pK_a_s for *C. eugametos* Hb are pH 6.4 and 8.5, the lower pKa assigned to a transition from pentacoordinate to partial hexacoordinate and the higher pKa of partial to full hexacoordinate ligation [38]. Also observed with the *C. eugametos* Hb was a partial change from penta- to hexacoordination upon reduction of the protein with dithionite. The Hb from *C. eugametos* is expressed in the nM range in chloroplasts in response to light and requires photosynthesis for its full expression [38,43,44].

### 3.1. General Model for the Effect of pH on the Coordination State of the Heme Iron

The effect of heme iron coordination on exogenous ligand binding is well known, with a model for the heme coordination of hexacoordinated globins and ligand binding kinetics previously described in detail [26,45,46] and summarized in Figure 1. In this scheme *P* is the protein, the suffix *h* and *p* denote hexacoordinate and pentacoordinate and *L* is the ligand. The rate constants for the forward and back reactions for the conformational change between the coordination states are denoted *k_f_* and *k_b_* and *k_L_[L]* the ligand concentration-dependent rate constant for ligand binding.

It is evident that on changing pH, HGb I undergoes a single proton-linked conformational change. Therefore, in the case of HGb-I, the model in Figure 1 is insufficient, and an extended model is required to take this effect of pH into account. The protonation-linked conformational change can happen in one of two ways: either protonation/deprotonation triggers dissociation of the ligand occupying the sixth coordination site to render the heme HS, or a minor, pre-existing, population of the HS form is stabilized by protonation and thus perturbs the equilibrium in favor of this form. A more detailed examination of these two models is given in the Appendix A. Fortunately, however, the pH jump data shown in Figure 3 provides a clear answer as to which mechanism prevails. The mechanism in Figure 2 is that in which protonation triggers conformational change. In this scheme, *H* is the proton and the equilibrium constant for proton dissociation is *K_a_*.

In the absence of an exogenous ligand and provided the deprotonation/protonation steps are fast compared with the conformational rate constant, this model yields the following expression for the single observed rate constant, k_obs_:k_obs_ = k_f_/(1 + K_a_/[H^+^]) + k_b_(1)

This expression predicts that k_obs_ approaches k_b_ at high pH values and (k_f_ + k_b_) at low pH values; i.e., it increases with decreasing pH. This transition follows a simple titration curve, with the mid-point yielding the pK of the triggering group, pK_a_. The alternative model predicts exactly the opposite behavior, namely a decrease in k_obs_ on increasing pH. Given the data in Figure 3A, we can confidently adopt the model in the scheme where it can be seen that a reasonable fit to Equation (1) can be obtained with a pKa ~ 3.0 for ferric protein and ~2.1 for ferrous protein. From the data in Figure 3A and Equation (1), we can calculate the values of k_f_ and k_b_ to be 6.3 and 0.06 for ferric protein and 48.6 and 4.1 for ferrous protein, respectively.

Furthermore, analysis of the equilibrium properties of the model reveals that the experimentally determined value of K_a(obs)_, for the transition to the HS form (Figure 2A) may be written in terms of the constants describing the model, namely:K_a(obs)_ = K_a_/(1 + K_c_)(2)
where K_c_ = k_f_/k_b_. From this it follows that provided K_c_ > 1 (as here) then
pK_a(obs)_ = pK_a_ + log K_c_(3) As our titration gave pK_a(obs)_ = 5.5 (Figure 2A) and the pH jump data shows pK_a_~3.0 for ferric HGb-I, it follows that the value of K_c_ ~ 3 × 10^2^. This means that once the trigger group is protonated, then the conformational equilibrium strongly favors the pentacoordinated HS form of the protein, for the ferrous protein this value of K_c_ is much higher, as the titration gave pK_obs_ = 8.4 and the pH jump data shows pK_a_ ~2.1, then K_c_~2 × 10^6^.

If lysine (E10) is the ligand at the distal side and it is this ligand that dissociates in the hexa to penta transition on protonation then the question arises as to the site of protonation. It is unlikely that the amine group can be protonated while bound to the iron, and the alternative of protonation post-dissociation of lysine is inconsistent with the pH dependence of the kinetics reported in Figure 3. Thus, the ‘trigger group’ must be other than the lysine that acts as the heme ligand. This situation has some parallels with the alkaline transition in cytochrome c, in which dissociation of a proton from a group of hydrogen-bonded residues close to the heme, and its associated propionic acid groups, leads to removal of methionine from iron ligation [47]. If a similar situation pertains to HGb-I, then at present we are unable to identify this ‘trigger’. An alternative suggestion is that the proximal histidine ligand that is protonated while bound to the heme and that this leads to dissociation of this residue to leave the heme pentacoordinate, the lysine remaining bound. This proposal has some attractive features, in particular the pK of the transition reported here is similar to that reported for the protonation of the proximal histidine in ferrous Mb, pK = 3.45 [48]. However, for Mb the protonation of the proximal histidine leads to its dissociation to yield a tetra coordinate species, which is not the situation with HGb-I as, under this hypothesis, the distal lysine remains bound. While we perhaps may assume that protonation is very rapid, as is the case in Mb, the dissociation of the His in HGb-I is relatively slow as shown in Figure 3 unlike in Mb where dissociation is very rapid. This may indicate that the proposal for the protonation of the proximal histidine is the event that drives the hexa to penta transition is unsound. However, as there are very significant structural differences between Mb and HGb-I one may argue that these differences account for the different kinetics observed. For example, protein structural and steric constraints on the proximal side of heme and also the fact that in Mb His dissociation is from a pentacoordinate form to yield a tetra-coordinate product in contrast in HGb-I it is from a lysine bound hexacoordinate form to a lysine bound penta coordinate species. A stronger argument against protonation of the proximal ligand leading to its dissociation may be found in the work of Tang et al. where it is shown that, in Mb, protonation of the proximal histidine does not lead directly to its dissociation to the heme [49].

### 3.2. Extension of the Model to Account for the Effect of Lipid Binding on the Coordination State of the Heme Iron

Here we attempt to bring together the pH and lipid induced heme coordination changes seen in HGb-I into the framework of a single simple model that can account for our experimental findings and which can provide explanations for the temperature independence of the binding constant for lipid.

From Figure 2A it is apparent that lipid binding and protonation are coupled, i.e., the pK_a(obs)_ of the LS to HS transition is moved to higher values in the presence of lipid. The model in Figure 1 may be extended to account for the observation that at a given pH value addition of lipid shifts the spin state equilibrium in favor of the HS, pentacoordinate form. A model to account for the coupling between lipid binding and conformational equilibria must involve a thermodynamic square in which lipid binds to all species and on binding triggers a conformational change. This model may be simplified, however, by proposing that the HS form of the protein has a much higher affinity for the lipid than the LS species. This is a reasonable proposal suggesting that dissociation of the intrinsic ligand from the sixth coordination site allows movement of the peptide chain that contains the ligand and thereby creates a site into which the lipid may insert. In this way the pentacoordinate form is stabilized. The model in Figure 2 now becomes Figure 3:

More complex models can be conceived, however this model adequately fits the observations and thus further complexities are unnecessary. It may be appreciated that within this model the apparent pK_a(obs)_ depends on lipid concentration and, reciprocally, the experimentally determined value of the equilibrium dissociation constant of the lipid K_a(obs)_ depends on the lipid concentration.

Analysis of this model provides equations that make these dependences explicit,

The K_a(obs)_ now has the following form
K_a(obs)_ = K_a_/(1 + K_c_(1 + [L]/K_L_))(4)

Inspection of this equation shows that it collapses to Equation (2) in the absence of added lipid.

We now see that pK_a(obs)_ depends on [L] such that the higher [L] becomes, the higher (more alkaline) is the measured pK_a(obs)_. Under conditions where [L] >> K_L_ then pK_a(obs)_ is directly proportional to log[L]. This equation also provides a way to calculate the intrinsic value of K_L_ by comparing the values of pK_a(obs)_ in the absence and presence of known lipid concentration.

Similar analysis shows that the experimentally determined value of the lipid dissociation equilibrium constant, K_L(obs)_ take the following form:(5)$$K_{L(obs)}=K_{L}(1+\frac{1}{K_{c}}(1+\frac{K_{a}}{\left[H\right]}))$$ Inspection of this equation shows that K_L_ increases with increasing pH and that at pH values above pH 7 this equation collapses to give the following approximation
(6)$$K_{L(obs)}=\frac{K_{L}}{K_{c}}(\frac{K_{a}}{\left[H\right]})$$

As it is expected that the dissociation constants, K_L_ and K_a_, both increase with temperature and as we see from Figure 6 and Appendix A that K_c_ also increases with temperature (i.e., the HS form is favored at higher temperatures) thus Equations (5) and (6) indicate that the effect of temperature tends to cancel leaving K_L(obs)_ little changed. This is seen in Figure 6.

Figure 3 also provides an explanation for the lipid concentration independence of the rate constant for lipid binding (Figure 7). At pH 7 the model proposes that lipid binds to the small population of protein in the HS form and thus perturbs the equilibrium in favor of this spin state. The rate constant for this is limited by the relaxation rate of the conformational equilibrium, which at pH 7 for the ferric protein is close to k_b_. An estimate for this constant may be obtained from fit to the pH jump experiments (Figure 3). These values are 6 × 10^−2^ and 4 s^−1^ for ferric and ferrous HGb I, respectively. Both of these values are ~20 fold higher than the 3.2 × 10^−3^ and 2.3 × 10^−1^ for dominant ferric and ferrous lipid binding rate constants given in Figure 7. However, the measured rate constant in Figure 7, although dependent on the value of k_b_, may be also affected by a pre-binding of lipid that enhances histidine dissociation, i.e., increased k_b_. Nevertheless, the data supports a higher rate constant for lipid binding to the ferrous protein compared to the ferric.

Given the value for pK_a(obs)_ in the presence of 25 µM lipid (8.40/9.24 ferric/ferrous, see Figure 2A) and the value for K_c_ (~3 × 10^2^/2 × 10^6^ ferric/ferrous), calculated from Equation (3), we may estimate the true value from the dissociation constant of the oleate from the pentacoordinate form of HGb-I. This calculation gives (K_L_) as ~30 nM and 4200 nM for ferric and ferrous oxidation states, respectively. Similarly, taking the measured value of K_L(obs)_ from the titration data obtained at pH 7 (Figure 6) and using Equation (5) we may calculate a value for K_L_ of ~20 nM for ferric and ~600 nM for ferrous protein. The two values for K_L_ obtained in this way clearly show that K_L_ is in the order of ~10^−7^ to 10^−8^ for the ferric protein and ~10^−6^ to 10^−7^ for the ferrous protein. Therefore, we may conclude that oleate binds very tightly to the protein in the pentacoordinate form, particularly in the ferric form of the protein.

### 3.3. Possible Role in pH Sensing for Lipid-Induced Coordination State Change

Globin-induced lipid oxidation for cell signaling, either under physiological or pathological conditions of oxidative stress, has have been proposed for Cygb and Mb, respectively [18,37,50]. With Mb, an acidic environment accelerates lipid oxidation reactions with alkalinization treatment of the tissue ameliorating the damaging effects of lipid oxidation [35]. However, the inability of HGb-I to generate lipid peroxides from phospholipids (Figure 8) essentially excludes the possibility of HGb-I generating oxidized lipid products for cell signaling. Hence, the purpose of the pH and lipid-induced change in heme iron coordination remains unclear. However, the shift in the pK_a(obs_) of the heme iron coordination may reflect a pH sensing mechanism, the details of which are at present unknown. The usually low redox potential of HGb-I (unligated −305 mV vs. SHE, Appendix A [5,18,28,51,52,53,54,55,56,57,58,59,60]) and its sensitivity to pH, being more significantly less negative at acidic pH (−191 mV, pH 3.5) [17], suggests that HGb-I does not act as an oxygen carrier but may act as an electron-transfer catalyst dependent on the FeIII/II redox pair of the heme group [17]. Acidophiles such as *M. infernorum* grow optimally at pH 2 to 2.5 [6]. However, the cytoplasm of such extremophiles is generally neutral or slightly acidic [61]. Although the cytoplasmic pH of *M. infernorum* has not been reported, similar acidophiles such as *Alicyclobacillus acidocalarius* maintain a cytoplasmic pH of between 6 and 7 when in media of pH 2 [61,62]. A high proton motive force across the bacterial cell membrane (in the order of approx. −250 mV) creates a high ΔpH and maintains pH homeostasis [61,63]. For *M. infernorum,* adaptations to an acidic environment may lie in the production of a series of enzymes. *Helicobacter pylori* generates ureases to neutralize protons entering the cell; however, *M. infernorum* does not have these ureases [6]. Instead, adaptations may include glutamate decarboxylase and arginine decarboxylase to counteract acidification of the cytoplasm by binding excess protons and releasing CO_2_ [6,64]. Another mechanism may involve agmatine hydrolysis by agmatine deiminase to release NH_3_, binding excess protons [65].

HGb-I is expressed in the cytoplasm; therefore, periplasmic pH sensing mechanisms likely do not involve HGb-I. However, the cytoplasmic membrane is the main barrier to proton influx [66]. Lipids are also important pH sensors, with pH changes inducing polarization and deformations of lipid bilayer assemblies [67]. Furthermore, environmental conditions can directly influence lipid biosynthesis in extremophiles [68]. This is vital for proton permeability adaptations to maintain membrane integrity at environmental extremes. When grown in acidic or high temperature conditions, non-extremophiles such as *Bacillus subtilis* restrict unsaturated fatty acid synthesis to alter the fluidity of its membranes, altering permeability [69]. Acidophiles can synthesize a highly impermeable membrane to respond to proton attack, composed of saturated and mono-unsaturated branched chain fatty acids [70]. Some thermoacidophiles including *A. acidocaldarius* contains unusual ω-alicyclic fatty acids as a major membrane component. Such lipids appear to be a key component of membranes to prevent acid damage form highly compact cell membranes [61,71,72]. Whatever the exact mechanism that counteracts acidification, the cytosolic pH of *M. infernorum* is likely around neutral pH, the pH range where maximum effect of lipid interaction is observed with ferric HGb-I, but not the ferrous protein (Figure 2). As lipids are an important part of pH sensing mechanisms there is the potential for HGb-I, processing a pH sensitive, temperature insensitive ability to change heme iron coordination in the presence of lipids, to be linked in this pH sensing mechanism. However, more studies are needed to confirm this and to evaluate the physiological relevance of the interaction of HGb-I with lipids and its relevance to potential pH sensing mechanisms.

## 4. Materials and Methods

### 4.1. Materials

Horse heart Mb, sodium oleate and all buffer materials were purchased from Sigma-Aldrich, Poole, UK. Isopropyl β-D-thiogalactopyranoside (IPTG) and 5-aminolaevulinic acid were from Molekula, Gillingham, UK. Carbon monoxide was purchased from BOC, Hadleigh, Suffolk UK.

### 4.2. HGb-I Expression and Purification

The gene for HGb-I was synthesized by Epoch Life Sciences Inc, Missouri City, TX, USA and incorporated into expression vector pET28a (Merck, Feltham, Middlesex, UK) such that the gene had an N terminal his-tag. Vector was transformed into BL21 DE3 cells (Sigma-Aldrich, Poole, Dorset, UK) by heat shock method (42 °C, 90 s). Protein was expressed in shaking flasks containing 1.4 L Luria-Bertoni media with kanamycin sulfate (50 µg mL^−1^), shaken at 120 rpm, 37 °C. When optical density was ~1.0 at 600 nm, protein expression was initiated by addition of 500 µM IPTG. Ferric citrate (50 μM) and 5-aminolaevulinic acid (250 μM) were added to the broth to augment heme synthesis. Carbon monoxide gas was bubbled through the solution for 30 s and the flasks were sealed and incubated for a further 18 h, shaken at 80 rpm, 37 °C. Cells were isolated by centrifugation (8000× *g*, 20 min), and cells were lysed using an Avestin Emulsiflex C3 homogenizer (Biopharma, Winchester, Hampshire, UK) at 15,000–20,000 psi, two passes. Following centrifugation (30,000× *g*, 20 min, 4 °C) supernatant was separated and buffer added to a final concentration of 20 mM sodium phosphate, 500 mM sodium chloride and 20 mM imidazole, pH 7.4. The his-tagged protein was purified using a GE Healthcare immobilized metal (nickel) affinity column (5 mL) and washed with phosphate buffer, 500 mM sodium chloride with 20 mM imidazole and eluted with phosphate buffer, 500 mM sodium chloride with 500 mM imidazole. Imidazole was removed by dialysis (1 mM sodium tetraborate, pH 9.5, 3 changes), and the his-tag was cleaved through incubation with bovine thrombin (Sigma-Aldrich; 10 units/mg of protein) at room temperature (22 °C) overnight with gentle mixing. Tag-free protein was purified using the nickel-affinity column (20 mM sodium phosphate, 500 mM sodium chloride and 20 mM imidazole, pH 7.4), followed by dialysis and concentration using a Whatman (Gillingham, Dorset, UK) 3 kDa spin filter.

### 4.3. Calculation of HGb-I Absorption Coefficient

Unknown concentrations of ferric HGb-I, along with a known concentration of Mb (calculated using the ferrous protein, molar absorption coefficient (ε435 nm = 121 mM^−1^ cm^−1^ [24]) were subjected to reverse-phase HPLC using an Agilent 1290 UPLC fitted with an Agilent 1260 diode array spectrophotometer (Stockport, Cheshire, UK). The column used was a Zorbax Stablebond 300C3 250 mm × 4.6 mm fitted with a 12 mm × 4.6 mm guard column, with a water/TFA—acetonitrile/TFA gradient as previously described [28,73]. The concentration of heme in the unknown HGb-I sample was calculated from the integrated area under the peak (~14 min elution time) and compared to the integral of the heme peak of known concentrations of Mb. This was repeated on three independent samples. The ε412 nm of ferric HGb-I, pH 7.4 was calculated to be 152 mM^−1^ cm^−1^ and was used for concentration determination in all experiments.

### 4.4. Lipid Titration and Lipid Binding Kinetics

Optical spectra were taken using an Agilent Cary 5000 spectrophotometer (Stockport, Cheshire, UK). Rapid reaction kinetics were performed on an Applied Photophysics SX20 stopped flow (Leatherhead, Surrey, UK) fitted with a diode array spectrophotometer and a Peltier temperature-controlled water bath. Rate constants were obtained from a global analysis using the Applied Photophysics ProK II software. Oleate binding follows a fractional saturation binding curse as previously described for Cygb [18,28]. The stoichiometry and dissociation constant of lipid binding to HGb-I was calculated from the following equation:(7)$$Y=\frac{\left({[P}_{T}]+\left[S\right]+K_{L}\right)-\sqrt{{\left({[P}_{T}]+\left[S\right]+K_{L}\right)}^{2}-4\left(\left[P_{T}\right]\left[S\right]\right)}}{2\left[P_{T}\right]}$$
where Y is the fractional saturation, [P_T_] is the total protein binding site concentration, [S] is the substrate concentration (oleate) and K_L_ is the lipid dissociation constant. Data were fitted to the equation using the least-squares method by either the Microsoft Excel solver program or KaleidaGraph v4.0 by Synergy Software.

### 4.5. Ligand Binding Kinetics

For CO binding kinetics, a solution of CO was prepared by the equilibration of water with one atmosphere of CO at 20 °C to give a solution of ~1 mM CO. This was transferred anaerobically to a glass syringe with an addition of a few grains of sodium dithionite to maintain an anaerobic environment. This was mixed in a 1:1 ratio with protein in 0.1 M buffer (pH 5–10, made anaerobic and deoxyferrous using sodium dithionite) on an Applied Photophysics SX20 stopped flow spectrophotometer. CO was diluted by mixing with anaerobic water using connected glass syringes. CO concentration was checked by mixing the CO with equine Mb and checking the rate constant for binding compared to previously reported rate constant [24]. For azide binding kinetics several solutions of azide (10 mM, 5 mM after mixing) was prepared in buffers pH 5–10 and mixed 1:1 ratio with ferric protein in 0.1 M buffer (pH 5–10) on an Applied Photophysics SX20 stopped flow spectrophotometer.

### 4.6. Electron Paramagnetic Resonance Spectroscopy

Protein aliquots (250 µL) were placed in Wilmad SQ EPR tubes (Stoke-On-Trent, Staffordshire, UK) and flash-frozen in dry ice-cooled methanol. Once the solutions were frozen, the tubes were wiped and transferred to liquid nitrogen. The electron paramagnetic resonance (EPR) spectra were measured at 10 K on a Bruker (Karlsruhe, Germany) EMX EPR spectrometer (X-band) at a modulation frequency of 100 kHz. A spherical high-quality Bruker resonator ER 4122 SP9703 and an Oxford instruments (Abingdon-on-Thames, Oxfordshire, UK) liquid helium system were used to measure the low-temperature EPR spectra. Baselines were corrected using WinEPR v. 2.22 (Bruker Analytik, GmbH) by subtraction of a polynomial line drawn through a set of points randomly chosen on the baseline. The relative concentrations of the HS and LS ferric heme forms were found from relative intensities of the g = 5.91 (for HS) and g = 3.05 (for LS) signals in the EPR spectra of samples with different proportions of LS and HS forms, given the same total ferric heme concentration in the two. Such an approach has been described for the quantitation of LS and HS ferric heme forms in Hb and Mb [74]. An EPR signal intensity in two different spectra can be measured in relative units with a high accuracy using the method of spectra subtraction with variable coefficient [75].

### 4.7. Lipid Oxidation Measurement

Liposomes were generated from soybean lecithin (Sigma, type II-S, P-5638) by initial sonication of lecithin granules (5 mg mL^−1^ in 0.1 M sodium phosphate buffer) until no granules were observed. Small unilamellar liposomes were then generated by extrusion through a 0.1 µm filter (ten passes), using a Northern Lipids extruder as previously described [37]. Liposomes were stored at 4 °C and used within 2 h of preparation. Lipid oxidation was monitored at 234 nm using an Agilent 8453 diode array spectrophotometer.

## 5. Conclusions

In summary, the phenomenon of HGb-I interaction with lipids to induce changes in the heme iron-coordination has many similarities with those observed in ferric human Cygb. However, the lack of lipid peroxidation activity of HGb-I and its ability to induce this change in heme coordination in the ferrous and ferric oxidation states suggest that any potential physiological role for the interaction of lipid is different from that of Cygb. Nonetheless changes on the affinity of the distal endogenous ligand as temperature changes is counteracted by modulation of lipid binding affinity, making the changes in heme iron coordination pH sensitive, but not temperature sensitive, lending weight to a potential pH sensing mechanism for HGb-I.

## Data Availability

Data are contained within the article and Appendix A.

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
