# Peer review of "Hell’s Gate Globin-I from *Methylacidiphilum infernorum* Displays a Unique Temperature-Independent pH Sensing Mechanism Utililized a Lipid-Induced Conformational Change"

_ijms, 2024, doi:10.3390/ijms25126794_

Round 1

Reviewer 1 Report

Comments and Suggestions for Authors

This great paper describes comprehensive coverage of Hell’ Gate globin-I. The experiments are well designed and the results are solid and reliable. The manuscript is well written and readable. I think that this interesting work would certainly advance our understanding of the structure and function relationships of HGb-I as well as those of other heme proteins. I found that this article is extremely interesting and thus definitely worthy for publication. I strongly recommend this paper for publication in the Journal. However, I raise some concerns that need to be addressed before publication. If those concerns are adequately addressed in the revised manuscript, this interesting report would be significantly strengthened.

Minor concerns that need to be addressed before publication.

[1] At the beginning of Introduction, short description about general roles of heme and structure and function relationships of Hb might be useful for general readers to start reading this paper. Direct introduction of HGb-I at the beginning of Introduction without those explanation in this manuscript would confuse non-experts and make them loose interests in this great study.  

[2] “Hell’s Gate globin-I” is often used, after HGb-I has been defined at line 26. For examples, please see lines 88, 100, 226, 287, 293, 299, 309, 321, 322, 324, 332, 341, 355, 362, 662, 663, 664 and others and also in Figure Legends in Supporting Information. Please remedy those.

[3] The authors often compare various factors of HGb-I with those of other heme proteins in the text. I would propose the authors to make a new table (in Supporting Information or on the main text) containing various factors including lipid binding, lipid peroxidase activity, O2 affinity, spin state, coordination structure, pH effect, temperature, redox potential et al. of HGb-I (in the presence and absence lipids), HGb-IV, Cygb, Ngb, Mb and other related hemeproteins in order to emphasize/contrast important differences between HGb-I and other heme proteins. 

[4] Additional figure (in Supporting Information or on the main text) listing amino acid sequence of those globin is suggested. Emphasis on differences in sequence among those globin proteins and on amino acid residues involving the heme coordination and heme surrounding might also be beneficial for general readers to grasp the points of the structure of HGb-I (in aid of lines 30-61 and others). 

[5] Is 3D structure including heme coordination structure available for HGb-I? Protein structure of HGb-I (in Supporting Information or on the main text) focusing on the heme active site would also useful for general readers, if it is available (in aid of lines 40-86 and others). If not available, please use/incorporate putative protein structure conjectured from those of other globin proteins. 

Please incorporate notes of E7-Gln, B10-Tyr (lines 48, 52, 168, 378), GH loop, H-helix (lines 56-57), E10-Lys (line s168, 382, 387) into this structure.  Notes of amino acids as heme axial ligands and amino acids interacting with the heme plain would also beneficial for readers. Please cite this figure, when you discuss those amino acid residues. 

[6] Legend for Figure 1, line 295: Dotted line should be “Dithionite-reduced ferrous form bound with CO”. Otherwise CO appears to be dithionite reduced as it is.

[7] Coordination structures: Incorporation of coordination structures of penta-coordinate, hexa-coordinate, ferrous, ferric heme complexes into Supporting Information or into the main text might be useful for general readers (in aid of lines 100-116 and others). 

[8] LS and HS appear at line 112, before their definition at line 119 and at line??? (where is definition of low-spin?).

[9] Legend for Figure 4, lines 324- 326 (in aid of lines 184-224): It seems strange “(C) Ferric HGb-I (5 μM) was …in the presence of sodium dithionite”.  I would suggest to use “(C) Dithionite-reduced ferrous HGb-I (5 μM) was titrated with sodium oleate…..pH 8.5”. 

[10] Legend for Figure 6, line 351: The same as [8]. Please use “… pH 8.5 for dithionite (20 mM)-reduced ferrous protein.” 

[11] Line 374: Should be (His-Fe(III)-His to His-Fe(III)-H2O) or (His-Fe3+-His to His-Fe3+-H2O).

[12] Lines 377-470: Please cite/use the additional new figure describing the protein structure and heme coordination structures as suggested in [4] above.

[13] Lines 400-489, Schemes 1, 2 and 3:  It is very confusing that H is used for hexacoordinate, P is used for pentacoordinate, and L is used for ligand here. This is because up to this section, H is used for high spin and L is used for low spin elsewhere in the text.  I would strongly suggest the authors to consider to use other abbreviations for Schemes 1, 2 and 3 and related sentences on this and following pages. This remediation is very important!

[14] Lines 598-608: Pyridine hemechromogen assay is usually used to determine heme concentrations and molar coefficients. Why did the authors not use this method?  For example, please see Bio Protoc. 2015, 5 (18) e1594. 

[15] Line 624: How do the authors determine the concentration of 1 mM CO? 

[16] Refs. 7, 11, 25, 39, and 43.   No doi e-numbers are described. Those are JBC papers, but JBC papers have the numbers as shown in Ref. 34

[17] I would suggest the authors to make a list of abbreviations at the beginning of the text or the end of Conclusion. However, the list might be omitted according to the journal policy. Please consult with the Editor about the list.

To sum up, if those minor concerns are adequately addressed in the revised manuscript, the present excellent paper would be further improved. I strongly recommend publication of this superb work after revisions are made.

Incidentally, heme-based oxygen sensors and heme-responsive sensors are emerging, as novel roles of hemes, as described in Chem. Rev. (IF 62.1) 2015, 115, 6491 and Chem. Soc. Rev. (IF 46.2) 2019, 48, 24, 5619, respectively, although the authors do not need cite those articles.

Author Response

Reviewer 1:

This great paper describes comprehensive coverage of Hell’ Gate globin-I. The experiments are well designed and the results are solid and reliable. The manuscript is well written and readable. I think that this interesting work would certainly advance our understanding of the structure and function relationships of HGb-I as well as those of other heme proteins. I found that this article is extremely interesting and thus definitely worthy for publication. I strongly recommend this paper for publication in the Journal. However, I raise some concerns that need to be addressed before publication. If those concerns are adequately addressed in the revised manuscript, this interesting report would be significantly strengthened.

We thank the reviewer their work and for their kind words. We have addressed the minor issues below:

Minor concerns that need to be addressed before publication.

[1] At the beginning of Introduction, short description about general roles of heme and structure and function relationships of Hb might be useful for general readers to start reading this paper. Direct introduction of HGb-I at the beginning of Introduction without those explanation in this manuscript would confuse non-experts and make them loose interests in this great study.  

Introductory passage added. 

[2] “Hell’s Gate globin-I” is often used, after HGb-I has been defined at line 26. For examples, please see lines 88, 100, 226, 287, 293, 299, 309, 321, 322, 324, 332, 341, 355, 362, 662, 663, 664 and others and also in Figure Legends in Supporting Information. Please remedy those.

Style has been changed to include abbreviations in figure titles and subheadings.

[3] The authors often compare various factors of HGb-I with those of other heme proteins in the text. I would propose the authors to make a new table (in Supporting Information or on the main text) containing various factors including lipid binding, lipid peroxidase activity, O2 affinity, spin state, coordination structure, pH effect, temperature, redox potential et al. of HGb-I (in the presence and absence lipids), HGb-IV, Cygb, Ngb, Mb and other related hemeproteins in order to emphasize/contrast important differences between HGb-I and other heme proteins. 

We thank the reviewer for the suggestion, although the relevant data for HGb-IV appears not to have been published, a table has been added with the majority of the requested data in table S1.

[4] Additional figure (in Supporting Information or on the main text) listing amino acid sequence of those globin is suggested. Emphasis on differences in sequence among those globin proteins and on amino acid residues involving the heme coordination and heme surrounding might also be beneficial for general readers to grasp the points of the structure of HGb-I (in aid of lines 30-61 and others). 

Amino acid sequence alignment of HGb-I and several other vertebrate and bacterial globins are presented in a new Figure S5. We have emphasized the amino acids discussed in the manuscript.

[5] Is 3D structure including heme coordination structure available for HGb-I? Protein structure of HGb-I (in Supporting Information or on the main text) focusing on the heme active site would also useful for general readers, if it is available (in aid of lines 40-86 and others). If not available, please use/incorporate putative protein structure conjectured from those of other globin proteins. 

Structures for HGb-I are available 3S1I, 3UBV and 3UBC3 (with oxygen bound) and S1J (with acetate bound). However, there are no ligand-free structures to show the hexacoordinate form, so the structure does not give much useful information beyond that of the sequence alignment and are therefore we believe inclusion of these structures will be more confusing than beneficial to the reader.

Please incorporate notes of E7-Gln, B10-Tyr (lines 48, 52, 168, 378), GH loop, H-helix (lines 56-57), E10-Lys (line s168, 382, 387) into this structure.  Notes of amino acids as heme axial ligands and amino acids interacting with the heme plain would also beneficial for readers. Please cite this figure, when you discuss those amino acid residues. 

As for point [5] we have not included 3d structures, however, key amino acids have been highlighted in Figure S5 sequence alignment.

[6] Legend for Figure 1, line 295: Dotted line should be “Dithionite-reduced ferrous form bound with CO”. Otherwise CO appears to be dithionite reduced as it is.

 Corrected.

[7] Coordination structures: Incorporation of coordination structures of penta-coordinate, hexa-coordinate, ferrous, ferric heme complexes into Supporting Information or into the main text might be useful for general readers (in aid of lines 100-116 and others). 

As detailed in point [5] the structures for HGb-I in the non-ligated state are not available. Without these structures, we believe that such a figure examining other globin would not be helpful.

[8] LS and HS appear at line 112, before their definition at line 119 and at line??? (where is definition of low-spin?).

Abbreviation now correctly defined at first use. The low- and high spin terminology is commonly used in the literature and therefore does not need definition.

[9] Legend for Figure 4, lines 324- 326 (in aid of lines 184-224): It seems strange “(C) Ferric HGb-I (5 μM) was …in the presence of sodium dithionite”.  I would suggest to use “(C) Dithionite-reduced ferrous HGb-I (5 μM) was titrated with sodium oleate…..pH 8.5”. 

The protein should have been stated as ferrous instead of ferric, which we have corrected. We have changed the terminology to state that the titration was done in the presence of “~20 mM sodium dithionite”. We feel it important to point this out as the protein is kept in the deoxygenated form during the oleate additions by the use of excess dithionite. Using ‘dithionite-reduced’, while emphasizing that the protein is ferrous, may not convey that the system was also kept deoxygenated throughout. 

[10] Legend for Figure 6, line 351: The same as [8]. Please use “… pH 8.5 for dithionite (20 mM)-reduced ferrous protein.” 

See response to [9].

[11] Line 374: Should be (His-Fe(III)-His to His-Fe(III)-H2O) or (His-Fe3+-His to His-Fe3+-H2O).

We thank the reviewer for pointing out this issue. Nomenclature has been adjusted to reviewer’s first example.

[12] Lines 377-470: Please cite/use the additional new figure describing the protein structure and heme coordination structures as suggested in [4] above.

References to S5 figure added to the main text.

[13] Lines 400-489, Schemes 1, 2 and 3:  It is very confusing that H is used for hexacoordinate, P is used for pentacoordinate, and L is used for ligand here. This is because up to this section, H is used for high spin and L is used for low spin elsewhere in the text.  I would strongly suggest the authors to consider to use other abbreviations for Schemes 1, 2 and 3 and related sentences on this and following pages. This remediation is very important!

There has been some misunderstanding as LS and HS are the abbreviations used for low spin and high spin (and then only in Scheme 1 and 2), not L and H. We have removed the references to spin state in the schemes to simplify and we have adjusted the schemes in the main text to reflect the nomenclature in the supporting information equations and schemes. This represents h and p (lower case subscript) for hexa/penta-coordination and H, P, and L (upper case) for proton, protein and lipid respectively. We hope this makes following the schemes and equations clearer.

[14] Lines 598-608: Pyridine hemechromogen assay is usually used to determine heme concentrations and molar coefficients. Why did the authors not use this method?  For example, please see Bio Protoc. 2015, 5 (18) e1594. 

The HPLC method is a simple one and comparable to the pyridine hemochromogen assay. It essentially uses the same principle of acidification (TFA instead of HCl) and separation using a solvent (acetonitrile instead of pyridine). Additionally as it cuts out extensive manual handling of the sample and automates concentration determination via integration of the heme peak, the results tend to be just as accurate as the pyridine hemochromagen assay, if not more so.

[15] Line 624: How do the authors determine the concentration of 1 mM CO? 

The Bunsen solubility coefficient gives the solubility of CO in the buffer at normal pressure as 1 mM. This has been checked by measuring the rate constant for binding with equine Mb. The text has been altered to clarify this.

[16] Refs. 7, 11, 25, 39, and 43.   No doi e-numbers are described. Those are JBC papers, but JBC papers have the numbers as shown in Ref. 34

Thank you for point these out. DOI’s have been added.

[17] I would suggest the authors to make a list of abbreviations at the beginning of the text or the end of Conclusion. However, the list might be omitted according to the journal policy. Please consult with the Editor about the list.

Through examination of other recently published articles, this journal does not appear to include abbreviation lists. However, we will verify this with the editor.

Reviewer 2 Report

Comments and Suggestions for Authors

The paper describes some thermodynamic and kinetic properties of HGb-1, a globin from an acidophile and thermophile M. infernorum. The data concern binding of lipids as a function of pH and temperature, which are interpreted in terms of possible pH sensing functionality.

The UV-vis and EPR spectroscopy and the kinetic experiments and analysis are properly done. Their interpretation in terms of functionality is somewhat mystifying. The EPR quantitation requires technical explanation.

(1) The main conclusion of the paper is that HGb-1 may act as a pH sensor. The context of this conclusion is not clear. The wiki of M. infernorum tells that it belongs to the phylum Verrucomicrobiota, which are Gram-negative bacteria. So M. infernorum has a plasma membrane and an outer membrane with a periplasmic space in between. To sense environmental pH, HGb-1 must first be exported to the periplasmic space. This aspect is completely ignored by the authors. To make sense out of the pH sensing, the sensor must also cause an effect by reacting with some cellular component. Also this aspect is completely ignored by the authors. What is the signal cascade? The authors conclude that HGb-1 presumably is an electron-transfer catalyst (line 548). What would be the function of this activity, and how does it relate to its proposed sensing function?

(2) The EPR spectra exhibit a low-spin signal of the HALS (highly anisotropic low spin) type for which only one g-value is observed. The authors refer to their own previous work (REF-27) to explain this type of signal. The reference is inappropriate: Ref-27 is about cytochrome c (not a HALS), for which two g-values are observed, and the third is not due to low protein concentration. In the present case, however, the third g-value is not observable for fundamental reasons. The authors claim that quantitation of the spectrum gives 91% of protein concentration. The suggested accuracy is quite unbelievable in view of the difficulty of quantifying these types of signals. The authors do not tell how they actually did the quantitation. Instead of their self-citing REF-27, they should have cited the classical 1979 paper by De Vries and Albracht [Biochim. Biophys. Acta 546: 334] in which it is explained how one can estimate HALS concentration based on a single peak. The EPR spectra also exhibit a high-spin signal. In order to quantitate such a signal, one needs information on the magnitude of the zero-field splitting. The authors ignore this complication. Please explain how the quantitation was done.

Author Response

Reviewer 2:

The paper describes some thermodynamic and kinetic properties of HGb-1, a globin from an acidophile and thermophile M. infernorum. The data concern binding of lipids as a function of pH and temperature, which are interpreted in terms of possible pH sensing functionality.

The UV-vis and EPR spectroscopy and the kinetic experiments and analysis are properly done. Their interpretation in terms of functionality is somewhat mystifying. The EPR quantitation requires technical explanation.

(1) The main conclusion of the paper is that HGb-1 may act as a pH sensor. The context of this conclusion is not clear. The wiki of M. infernorum tells that it belongs to the phylum Verrucomicrobiota, which are Gram-negative bacteria. So M. infernorum has a plasma membrane and an outer membrane with a periplasmic space in between. To sense environmental pH, HGb-1 must first be exported to the periplasmic space. This aspect is completely ignored by the authors. To make sense out of the pH sensing, the sensor must also cause an effect by reacting with some cellular component. Also this aspect is completely ignored by the authors. What is the signal cascade? The authors conclude that HGb-1 presumably is an electron-transfer catalyst (line 548). What would be the function of this activity, and how does it relate to its proposed sensing function?

We thank the reviewer for these points in detailing potential mechanisms of pH sensing. We take on board that there are diverse pH sensing mechanisms for both the cytoplasm and periplasmic space. However, changes in the cytoplasmic cell membrane composition appear to play a key role in preventing acid attack of the bacterium. We added referenced text to the pH sensing subsection of the discussion to emphasise this. Hence our proposal is for a potential pH sensing in the cytoplasm as it is critical for preserving a neutral or slightly acidic pH to maintain function in the cytoplasm. We do not claim that HGb-I is pH sensing in the periplasm or is transported to the periplasmic space as there is no data or publication to our knowledge supporting the presence of the globin outside the cytoplasm. We have added text to clarify this.

We have also added to the discussion to highlight the current knowledge and importance of lipids as pH sensors and the effect of pH on the lipid composition of acidophiles and to present a potential pathway to link between our observations that lipid-binding may be linked to pH sensitive, temperature insensitive changes to HGb-I chemistry. We acknowledge, however, that this hypothesis requires further exploration to validate. We hope this is sufficient to alleviate the reviewers concerns.

(2) The EPR spectra exhibit a low-spin signal of the HALS (highly anisotropic low spin) type for which only one g-value is observed. The authors refer to their own previous work (REF-27) to explain this type of signal. The reference is inappropriate: Ref-27 is about cytochrome c (not a HALS), for which two g-values are observed, and the third is not due to low protein concentration. In the present case, however, the third g-value is not observable for fundamental reasons.

We have made some edits to the manuscript, in section 2.1.4. We added references to the de Vries and Aasa & Vanngard papers. We have kept ref 27 (now ref 33) as the gz there is 3.05, which is >3, and thus qualified for de Vriese’s definition of highly anisotropic low-spin. We did not say that the third component is not observed due to low protein concentration, so the Aasa/Vanngard reasoning is applicable to all.

The authors claim that quantitation of the spectrum gives 91% of protein concentration. The suggested accuracy is quite unbelievable in view of the difficulty of quantifying these types of signals. The authors do not tell how they actually did the quantitation. Instead of their self-citing REF-27, they should have cited the classical 1979 paper by De Vries and Albracht [Biochim. Biophys. Acta 546: 334] in which it is explained how one can estimate HALS concentration based on a single peak. The EPR spectra also exhibit a high-spin signal. In order to quantitate such a signal, one needs information on the magnitude of the zero-field splitting. The authors ignore this complication. Please explain how the quantitation was done.

We thank the reviewer for picking up on this point: the way we determined the concentrations of the paramagnetic centers should have been given in the Methods section. We now made an addition to the section.  The accuracy we report is quite achievable for the EPR signals with undetermined line-shapes – all is needed is any two samples with these two EPR signals admixed at different proportions.  Just relative intensities of a component in each signal (g=6 of the HS signal and g=3 of the LS signal), and the knowledge that the two forms in total equate to the same heme concentration, allow to compose an equation with one unknown. (Journal of Magnetic Resonance (2000) 142 (2), 266-275).